# Novel Putative *Tymoviridae*-like Virus Isolated from *Culex* Mosquitoes in Colombia

**DOI:** 10.3390/v15040953

**Published:** 2023-04-13

**Authors:** Katherine Laiton-Donato, Camila Guzmán, Erik Perdomo-Balaguera, Ladys Sarmiento, Orlando Torres-Fernandez, Héctor Alejandro Ruiz, Alicia Rosales-Munar, Dioselina Peláez-Carvajal, Maria-Cristina Navas, Matthew C. Wong, Sandra Junglen, Nadim J. Ajami, Gabriel Parra-Henao, José A. Usme-Ciro

**Affiliations:** 1CIST—Centro de Investigación en Salud para el Trópico, Facultad de Medicina, Universidad Cooperativa de Colombia, Santa Marta 470003, Colombia; 2Grupo de Virología, Dirección de Redes en Salud Pública, Instituto Nacional de Salud, Bogota 111321, Colombia; 3Grupo Genómica de Microorganismos Emergentes, Dirección de Investigación en Salud Pública, Instituto Nacional de Salud, Bogota 111321, Colombia; 4Secretaría de Salud Distrital, Programa de Enfermedades Transmitidas por Vectores, Santa Marta 470004, Colombia; 5Grupo de Morfología Celular, Dirección de Investigación en Salud Pública, Instituto Nacional de Salud, Bogota 111321, Colombia; 6Grupo de Gastrohepatología, Facultad de Medicina, Universidad de Antioquia, Medellin 050010, Colombia; 7Platform for Innovative Microbiome and Translational Research (PRIME-TR), Department of Genomic Medicine, The University of Texas MD Anderson Cancer Center, Houston, TX 77030, USA; 8Viroworks, Houston, TX 77030, USA; 9Institute of Virology, Charité Universitätsmedizin Berlin, Corporate Member of Free University Berlin, Humboldt-University Berlin, and Berlin Institute of Health, 10117 Berlin, Germany

**Keywords:** virus discovery, molecular characterization, *Tymoviridae*, *Guachaca virus*, next-generation sequencing

## Abstract

The family *Tymoviridae* comprises positive-sense RNA viruses, which mainly infect plants. Recently, a few *Tymoviridae*-like viruses have been found in mosquitoes, which feed on vertebrate sources. We describe a novel *Tymoviridae*-like virus, putatively named, *Guachaca virus* (GUAV), isolated from *Culex pipiens* and *Culex quinquefasciatus* species of mosquitoes and collected in the rural area of Santa Marta, Colombia. After a cytopathic effect was observed in C6/36 cells, RNA was extracted and processed through the NetoVIR next-generation sequencing protocol, and data were analyzed through the VirMAP pipeline. Molecular and phenotypic characterization of the GUAV was achieved using a 5′/3′ RACE, transmission electron microscopy, amplification in vertebrate cells, and phylogenetic analysis. A cytopathic effect was observed in C6/36 cells three days post-infection. The GUAV genome was successfully assembled, and its polyadenylated 3′ end was corroborated. GUAV shared only 54.9% amino acid identity with its closest relative, *Ek Balam virus*, and was grouped with the latter and other unclassified insect-associated tymoviruses in a phylogenetic analysis. GUAV is a new member of a family previously described as comprising plant-infecting viruses, which seem to infect and replicate in mosquitoes. The sugar- and blood-feeding behavior of the *Culex* spp., implies a sustained contact with plants and vertebrates and justifies further studies to unravel the ecological scenario for transmission.

## 1. Introduction

The family *Tymoviridae* belongs to the order *Tymovirales*, a diverse group represented by viruses that infect plants. The family comprises three genera, *Tymovirus*, *Marafivirus*, and *Maculavirus,* which share their virus morphology with non-enveloped virions and icosahedral capsids of coat proteins that are approximately 30 nm in diameter [1]. These viruses are considered members of the alphavirus-like superfamily because of the genome organization, replication, and translation strategies [2]. The identified species contain positive-sense, single-stranded, and 5′capped RNA genomes of approximately 6–7.5  kb encoding at least a large polypeptide that is proteolytically processed by a virus-encoded endopeptidase into proteins with methyltransferase, NTPase/helicase, and RNA-dependent RNA polymerase activities, making up the replicase complex, and depending on the genus, a second protein (coat) translated from a subgenomic RNA at the 3′ region, and other overlapping proteins, such as the movement protein in *Tymovirus*, P16 and P31 in *Maculavirus*, and P21 in *Marafivirus* [1]. While the presence of a tRNA-like structure (TLS) is found in the genus *Tymovirus*, poly(A)-tailed genomes are characteristic of the genera *Maculavirus* and *Marafivirus* (except for *Maize rayado fino virus*) [3]. Their translation and RNA genome replication occur in the cytoplasm in virus-induced membrane structures at the periphery of chloroplasts and mitochondria [1,4].

Virus transmission in plants occurs through mechanical contact, infected plant material, or insect vectors [1,5]. Insect species from the families *Chrysomelidae*, *Curculionidae*, and *Cicadellidae* may act as vectors of these viruses [1]. While an ecological relationship between plant viruses and plant fluid-feeding insects can be directly inferred, and sustained virus exposure to the cellular environment of the insect vector could impose selection pressure for virus adaptation, there is little evidence of a biological role of insects in actively replicating and amplifying plant viruses at titers for a circulative and propagative transmission [6]. The male mosquito diet (and a complement for females) depends on obtaining sugars and other nutrients from plants [7], which is a plausible explanation for the increasing report of tymoviruses in insects detected through metagenomic studies [8,9,10,11,12,13,14,15,16]. Due to the sugar- and blood-feeding behavior of mosquitoes on plant and vertebrate sources, respectively, the recent isolation of *Tymoviridae*-like viruses in China, Mexico, and Brazil is of major interest [17,18,19], justifying studies of ecological interactions favoring transmission.

Here, we describe the discovery, using molecular and phenotypic characterization, of a new member of the family *Tymoviridae,* putatively named the *Guachaca virus* (GUAV), which was isolated from a mosquito pool of the genus *Culex* in the Sierra Nevada de Santa Marta (SNSM) and located in the north coast of Colombia, an ecosystem that comprises natural (sylvatic) and intervened (rural and urban) areas.

## 2. Materials and Methods

### 2.1. Study Area and Entomological Collection

The viral metagenomic study in hematophagous mosquitoes was carried out in the natural ecosystem of the Sierra Nevada de Santa Marta, Santa Marta, Magdalena, Colombia. The SNSM is an isolated mountain range in northern Colombia, which is separate from the Andes range that runs through the north of the country. Reaching an elevation of 5700 m (18,700 ft) just 42 km (26 mi) from the Caribbean coast, SNSM is the highest coastal range in the tropics, and one of the highest coastal ranges in the world. SNSM encompasses about 17,000 km^2^ (6600 sq mi) and serves as the source of 36 rivers. The range is in the Departments of Magdalena, Cesar, and La Guajira. A pool containing six specimens identified through classic taxonomy as belonging to the genus *Culex* (*Culex pipiens*, *Culex quinquefasciatus*) was collected on 8 June 2018, using a CO_2_-baited CDC light trap in a rural setting of Quebrada Valencia (latitude 1,124,300,000 and longitude −7,380,511,111) near the populated center of Puerto Nuevo-Guachaca area. The mosquito pool was transported in liquid nitrogen from the field station to the molecular biology laboratory at the Universidad Cooperativa de Colombia, where it was stored at −80 °C until processing.

### 2.2. Virus Isolation and Preparation of Viral Stocks

C6/36 cells (CRL-1660, obtained from ATCC), derived from whole *Aedes albopictus* larvae were cultured in Eagle’s minimal essential medium (MEM), supplemented with 10% and 2% fetal bovine serum (FBS) for growth and maintenance, respectively, and incubated in an atmosphere at 95% relative humidity and 5% CO_2_ at 28 °C. Vero cells (CCL-81, obtained from ATCC) derived from the kidney of the African green monkey *Cercopithecus aethiops* were cultured in MEM with 8% and 2% FBS for growing and maintenance, respectively, and incubated in an atmosphere at 95% relative humidity and 5% CO_2_ at 37 °C.

The mosquito pool was mechanically homogenized for 25 s and 1500 rpm in the BeadBug instrument (Benchmark Scientific Inc., Sayreville, NJ, USA) with the use of ceramic Magna lyser green beads (Roche, Mannheim, Germany) in 1.3 mL of Dulbecco’s phosphate-buffered saline (DPBS) supplemented with 10% of FBS and 1% of penicillin/streptomycin. Subsequently, it was centrifuged at 14,000 rpm for 15 min and filtered with the use of a 0.4 µm syringe filter. A 100 aliquot of the supernatant was used for inoculation on C6/36 and Vero cell cultures growing in 24-well plates, and the adhesion phase was carried out for 1 h, at 28 °C and 37 °C, respectively. Subsequently, 800 µL of MEM supplemented with 2% FBS and antibiotics were added, and the cultures were incubated at the appropriate temperature. Negative (mock infection) control was included in each multi-well plate. Supernatant of the virus isolation was collected on the third day post-infection when cytopathic effect was evidenced. Subsequently, a second passage was carried out in the same conditions. Each supernatant was collected, cleared, aliquoted, and stored at −70 °C.

Virus stocks were obtained from the inoculation of two million C6/36 cells in T25 flasks with a 1/10 dilution of the supernatant of the second passage virus isolation. The inoculation was carried out with 125 µL of the supernatant and 1125 µL of MEM with 2% SFB and 5% tryptose phosphate during one hour at 28 °C. Then, MEM was added to a final volume of 5 mL, and the cell culture was incubated until the cytopathic effect was observed in more than 50% of the cell monolayers around the first three days post-infection. The presence of the virus was confirmed by real-time RT-PCR using custom-made primers as described below.

### 2.3. RNA Purification

RNA purification from the supernatant of mosquito pool homogenization was performed using the QIAamp Viral RNA mini kit (Qiagen, Hilden, Germany), according to the manufacturer instructions, followed by RNase-free RQ1 DNase treatment to degrade double- and single-stranded DNA (Promega, Dübendorf, Switzerland). The RNA extract was stored at −80 °C.

### 2.4. Metagenomic Sequencing, Assembly, and Virus Discovery

The RNA extract was used for whole transcriptome amplification through the WTA2 kit (Merck KGaA, Darmstadt, Germany) following the NetoVIR protocol [20]. Subsequently, dsDNA was quantified by fluorometry, and 3 ng was used for library preparation with the Nextera XT DNA library prep kit (Illumina, San Diego, CA, USA). Next-generation sequencing (NGS) was performed in a MiSeq instrument with the MiSeq reagent kit version 3 of 600 cycles (Illumina, San Diego, CA, USA). Resulting sequencing reads were processed with VirMAP, which employs a combination of de novo and reference-guides assemblies and taxonomic classification using GenBank databases “gbvrl” and “gbphage” [21].

### 2.5. Primer Design for Virus Detection by RT-qPCR

Primer design was carried out for the consensus sequence assembled for the GUAV identified in the present study (Table 1) using the Geneious software v9.1.8 (Biomatters Ltd., San Diego, CA, USA). Real-time RT-PCR was carried out with annealing temperature of 55 °C and the SuperScript™ III Platinum One-Step qRT-PCR kit (Thermo Fisher Scientific, Carlsbad, CA, USA), according to the manufacturer instructions, in the CFX96 Real-Time PCR system (Bio-Rad Laboratories, Inc., Hercules, CA, USA).

### 2.6. 5′ and 3′ UTR Amplification Using RACE

For 5′ UTR amplification, the 5′/3′ RACE kit second generation (Roche Diagnostics GmbH, Mannheim, Germany) was used. After cDNA synthesis and poly(A) tailing, PCR amplification steps were performed to the poly(A)-tailed cDNA, using the oligo dT-anchor primer and the designed reverse primers, Tymoviridae-like5UTR-R2 and Tymoviridae-like5UTR-R1 (Table 1), according to the manufacturer’s instructions.

To corroborate the presence of the poly(A) tail at the 3′ end of the viral genome previously evidenced by the NGS virus assembly, an assay of cDNA synthesis was carried out with random hexamers or Oligo(dT) primer, followed by PCR amplification with the designed Scheme_20_RIGHT_2 (5′-ttacaaaccaggaacctctgact-3′) and the Oligo(dT) primers. The amplification was performed with final concentration of 250 nM of each primer, 1X PCR buffer, 2 mM MgCl_2_, 400 µM of each dNTP, 1.25 U of Platinum *Taq* DNA polymerase (Invitrogen, Carlsbad, CA, USA), and 5 µL of the polyadenylated cDNA in a final reaction volume of 25 µL. The thermal profile was as follows: 95°C for 4 min, 40 cycles (95 °C for 20 s, 55 °C for 30 s, and 72 °C for 30 s), and a final extension at 72 °C for 5 min. The purification of PCR amplicons was performed using the QIAquick PCR purification kit (Qiagen Inc., Chatsworth, CA, USA). DNA sequencing was performed through the Sanger method (Macrogen Inc., Seoul, Republic of Korea). The DNA sequences were edited and assembled into the Geneious software v9.1.8 (Biomatters Ltd., San Diego, CA, USA).

### 2.7. Complete Genome Annotation

Consensus sequences of the viral genome obtained from NGS and that of the 3′ UTR obtained by the RACE approach and direct sequencing were assembled through Geneious v9.1.8 (Biomatters Ltd., San Diego, CA, USA), and motifs, secondary structures, and antigenic domains were predicted and annotated. InterPro server from ELIXIR infrastructure was used for functional domain prediction [22].

### 2.8. Evaluation of Virus Growth in Mosquito and Permissiveness of Mammalian Cells

C6/36 cells growing in T25 flasks were inoculated with a 1/10 dilution of the viral stock and incubated under standard conditions, as described above. Supernatants were collected at 3-, 6-, 9-, 12-, 15-, 18-, 24-, 36-, 48-, and 72 h post-infection, and triplicates were evaluated using real-time RT-PCR. Cell cultures were examined every day for the presence and magnitude of cytopathic effect during the assay.

To assess the ability of GUAV to infect human cells, HEK293 (human embryonic kidney), HeLa (adenocarcinoma human cervix epithelial), A549 (adenocarcinoma human alveolar basal epithelial), and U937 (pro monocyte from histiocytic lymphoma) were cultured in T25 flasks at 37 °C with 5% CO_2_. Each cell line was grown in the following specific medium conditions: HEK293 in DMEM 10% FBS, HeLa and A549 in MEM 8% FBS, and U937 in RPMI 5% FBS and stimulated with PMA at final concentration of 25 µM. Confluent monolayers or suspensions were inoculated with 500 μL of a 1/10 dilution of the viral stock, following BSL-3 practices. The adhesion phase was carried out at 37 °C for 1 h and then each flask was rinsed twice with 5 mL of DPBS, and 5 mL of fresh culture medium supplemented with 2% FBS were added. Cell culture supernatants were collected at 5, 7, 9, 11, and 13 days post-infection, aliquoted, and stored at −70 °C for subsequent RNA purification and real-time RT-PCR.

### 2.9. Transmission Electron Microscopy (TEM)

C6/36 cells infected with the *Tymoviridae*-like virus were fixed in 3% glutaraldehyde prepared in phosphate buffer pH 7.2, centrifuged at 3000 rpm for 5 min, resuspended, and washed three times with the phosphate buffer pH 7.2. Subsequently, post-fixation with 1% osmium tetroxide was carried out for one hour, followed by three washes, as described above. The cells were dehydrated by treatment with ethanol solutions in ascending concentrations (50, 70, 80, 95, and 100%) for 15 min each, followed by propylene oxide for 20 min. For the pre-imbibition, the cell pellet was treated with a mixture of propylene oxide and resin Epon-Araldite in a 2:1 ratio followed by another 1:1 mixture, for one hour. To complete the imbibition, pure resin was added, and the cell pellet was left at 4 °C. The next day, the resin was replaced and the was transferred to an oven at 65 °C for 24 h to obtain the polymerized block. In an ultramicrotome, semi-fine sections (0.5 microns) and ultra-fine sections (60 nanometers) were obtained. The sections were contrasted with uranyl acetate and lead citrate for observation in an electron microscope EM109 (Zeiss, Jena, Germany).

### 2.10. Phylogenetic Analysis

A nucleotide sequence dataset was generated through the identification of the ORF1 in previously released sequences of members of the family *Tymoviridae*, followed by a multiple sequence alignment (MSA) at the nucleotide and amino acid levels using MAFFT v.7 [23]. Accounting for the high genetic distance between some species of the family, a well-represented and unambiguous region of 3480 nt was selected for phylogenetic analysis. The nucleotide substitution model was estimated for the dataset through ModelFinder [24] and the maximum likelihood method was used for tree reconstruction with IQ-TREE [25]. Branch support was estimated by UltraFast Bootstrap with 1000 replicates [26].

## 3. Results

### 3.1. Cytopathic Effect Is Suggestive of Productive Viral Infection in C6/36 Cells

C6/36 cells inoculated with a supernatant of a homogenized mosquito pool of *Culex* spp. (CIST0019) displayed a cytopathic effect that was characterized by the gradual detachment of the cell monolayer by the third day and expanded to the whole culture by the fifth day post-infection. A second passage of a 1/10 dilution of the cell supernatant in fresh C6/36 cells induced the same effect over the following three days, demonstrating the active replication of a transmissible agent (Figure 1).

### 3.2. A Tymoviridae-like Virus Identified through Metagenomic Next-Generation Sequencing

A total of 92,939 paired-end reads were obtained with a Q-score threshold of ≥30. Viral metagenomic analysis using VirMAP [21] enabled the identification of 81,989 viral reads. A total of 81,165 reads (99%) mapped to *Ek Balam virus* with a genome coverage of 96.2% and a depth of 1824X. Other viral species were identified in the same sample, accounting for an extremely low proportion: *Dipteran brevihamaparvovirus* (768 reads, 0.9%), *Rinkaby virus* (43 reads, 0.05%), *Culex flavivirus* (10 reads, 0.01%), and *Atrato Gouko-like virus* (3 reads, 0.004%). Low nucleotide and amino acid identities of 57.8% and 54.9%, respectively, to *Ek Balam virus* were observed, supporting its proposal as a novel species within the family *Tymoviridae*, putatively named the *Guachaca virus* (GUAV).

### 3.3. Putative Guachaca Virus Genome Organization

The full-genome sequence of the putative GUAV was reconstructed from the NGS data plus 3′ UTR corroboration (Figure 2a). The 5′ UTR amplification attempts were unsuccessful; however, the RNA secondary structure of the available sequence at the 5′ end allowed the prediction of a highly stable region (dG = −3819 kcal/mol) with a long and perfectly matched stem-loop structure of 60 nucleotides, similar to those found in other RNA viruses [27], resembling a viral internal ribosomal entry site (IRES) (Figure 2b) [28], and suggesting a selective pressure for a functional role [29]. Extending from the nucleotide position 154 to 5388, a large ORF encoding for a 1744 residues polyprotein was predicted (ORF1). The functional domain’s composition of this ORF was similar to other members of the alphavirus-like superfamily and members of the family *Tymoviridae*, with the conserved methyltransferase, endopeptidase, helicase, and RNA polymerase sequence motifs resembling a replicase complex (Appendix A). A second ORF (ORF2) was predicted from the nucleotide positions 5418 to 6149, with only 29 nucleotides downstream of the ORF1 stop codon, showing homology to the coat protein of several species belonging to the family *Tymoviridae*, and displaying a cleavage signal at the amino-terminal region and several antigenic domains distributed throughout the soluble portion of the polypeptide (Appendix A). At the 3′ UTR, a 104 nucleotides sequence was corroborated through RACE-PCR, which was followed by direct sequencing. This sequence was predicted to fold into a partially matched stem-loop structure, followed by a poly(A) tail of approximately 21 nucleotides (Appendix A). GUAV genome reconstruction and annotations are available under GenBank accession number: OQ286121.

### 3.4. Phylogenetic Inference Supports the Proposal of a New Virus Species and Suggest the Need for New Genera for Insect-Specific Viruses from the Family Tymoviridae

For phylogenetic analysis, an MSA of 2042 amino acids from a conserved region of the ORF1, ranging from amino acid positions 704 to 1723, was performed for 56 sequences, which were representative of the different genera within the family *Tymoviridae*, and five sequences of species belonging to the family *Alphaflexiviridae*, which were incorporated as an outgroup. The genera *Tymovirus* and *Marafivirus* were well represented, and most species belonging to these genera fell into corresponding monophyletic groups, except for *Poinsettia mosaic virus,* a species originally assigned the genus Tymovirus, which fell into the genus *Marafivirus* in our study, and *Alfalfa virus F* and *Medicago sativa marafivirus 1*, which were distantly related to the other species within the genus *Marafivirus* (Figure 3). The genus *Maculavirus* was represented by only two members of the same species (*Grapevine Red Globe virus*).

It is of special interest that an increasing number of recently described species, and especially those detected or isolated from insects, differ from the previously described genera. This well-supported monophyletic group contains at least three insect-associated *Tymoviridae* (IAT) lineages; one of them is represented by the proposed virus species infecting *Bombyx mori*, *Andrena haemorrhoa*, and *Lampyris noctiluca*, and the second and third IAT-lineages are represented by viruses infecting mosquitoes of the family Culicidae. The second IAT-lineage with a branch support of 100% comprised several proposed species, including the putative GUAV isolated in 2018 in Colombia from *Culex* spp. mosquitoes; *Ek Balam virus*, isolated in 2007 in Mexico from *Culex quinquefasciatus*; and *Mutum virus*, isolated in 2018 in Brazil from *Mansonia* sp. A third IAT-lineage of Culicidae-infecting viruses with a branch support of 100% comprised the unclassified *Guadeloupe Culex tymo-like virus*, *Tarnsjo virus*, *Tymoviridae* sp., *Sichuan mosquito tymo-like virus*, and *Culex pseudovishnui tymo-like virus*, detected or isolated from *Culex* spp., and unclassified mosquitoes from Japan, Guadeloupe, Sweden, and China (Figure 3). The evolutionary divergence estimated from the amino acid differences (p-distances) showed that within-lineage mean distances for the three suggested IAT-lineages were comparable to those obtained for the previously described genera (*Maculavirus*, *Marafivirus,* and *Tymovirus*), consistent with their monophyletic grouping (Table 2). The evolutionary divergence over the sequence pair between the groups showed lower values between the previously described genera (0.456–0.515) than between almost any of these genera and the suggested IAT-lineages (0.520–0.622), except for the *Marafivirus*-IAT_Lineage_3 (0.505) (Table 2).

### 3.5. Ultrastructural Features of Guachaca Virus

The TEM results showed complete icosahedral viral particles with a diameter of 40–50 nm in symmetrical arrangements in the cytoplasm, as well as associated with vesicles that were fused to the cell membrane in the cell periphery of C6/36 cells infected with GUAV 5 days post-infection (Figure 4).

### 3.6. Vertebrate Cells Were Not Permissible for GUAV Infection

While GUAV successfully replicated in C6/36 cells, showing an increasing titer during the first 72 hpi (Appendix A), this virus isolation was unable to replicate in vertebrate cells (Vero, HEK, HELA, A549, and U937). Cell cultures were observed every day without evidence of a cytopathic effect in vertebrate cells and without detection or accumulation of viral RNA by RT-qPCR (Appendix A).

## 4. Discussion

The genus *Culex* of mosquitoes (Diptera: Culicidae) is of special relevance for public health, with several species serving as vectors for critical human and animal pathogens (e.g., *Mayaro virus* (MAYV), SLEV, *West Nile virus* (WNV), etc.). The anthropophilic behavior of this vector species highlights its proximity with humans which, certainly leads to a sustained exposure to the vector virome and the potential transmission of pathogens.

After inoculation of a C6/36 cells culture with a supernatant of *Culex* spp. that were pool collected in the Sierra Nevada de Santa Marta, an isolated mountain range in the north of Colombia, the cytopathic effect was observed from 3 dpi, gradually extended to the whole monolayer, and was reproduced after a second passage of a diluted inoculum. This finding suggested the presence of a transmissible agent, which was further studied and proposed as a new virus species.

Metagenomic approaches for virus discovery have been refined during the last decade, with the publication of several experimental protocols for virus enrichment and pipelines for taxonomic assignment. In our study, the VirMAP pipeline was successfully applied, allowing the identification of a viral contig that is distantly related to all previously reported viruses, but most closely related to the *Ek Balam virus.* The metagenomic approach is powerful for the identification of viral signatures, whose genome coverage depends on the viral abundance in a sample. Our results were successful in obtaining the complete genome with high coverage of a new member of the family *Tymoviridae,* putatively named the *Guachaca virus* (GUAV). Similar findings have been previously reported in China, Mexico, and Brazil [17,18,19], where genetically related members of the family *Tymoviridae* were isolated from *Culex* spp., suggesting the existence of an unclassified group of virus species in this family.

The Inference of the genome organization of GUAV demonstrated that this virus differed from the three previously defined genera, with the absence of a tRNA-like structure and a movement protein characteristic of the genus *Tymovirus*, and the absence of the accessory proteins present in the genera *Maculavirus* and *Marafivirus*.

The RACE strategy for the 5′ end characterization of the GUAV genome was unsuccessful, but the genome assembly from the NGS data enabled the prediction of secondary RNA structures in that region, starting with a long, perfectly matching stem-loop, followed by a structure of major complexity. The RACE strategy applied to the 3′ end enabled the corroboration of the presence of a poly(A) tail. It is of interest that a highly structured 3′ end with a long stem-loop and internal bulges was successfully identified upstream of the poly(A) tail. Tymoviruses have explored several solutions for the 3′ end stability and function, including unstructured polyadenylated and non-polyadenylated tails, tRNA-like structures [3], and the here described structured and polyadenylated end.

While the family *Tymoviridae* is mainly represented by plant-infecting viruses, the phylogenetic analysis presented here and in some previously reported studies [17,18,19], support the emergence of monophyletic groups conformed by insect-infecting viruses, which could justify the need for new genera proposals and reclassification of the virus species inside the family.

With the design of a molecular approach, the productive infection in C6/36 cells was demonstrated as an increase in viral genome abundance, which was inversely proportional to the Ct values that gradually decreased between the first- and third-day post-infection. This evidence of infection in mosquito cells indirectly suggests that GUAV could be infecting natural populations of *Culex* mosquitoes, which is also supported by previous studies with closely related viruses [18,19,30].

The TEM findings evidenced the presence of virus-like particles in the cytoplasm of C6/36 cells and particles associated with intracellular membranes, as has been evidenced for other members of the family *Tymoviridae* in which peripheral vesicles in chloroplasts and/or mitochondria have been reported [1].

All tested mammalian cells were not permissive for GUAV replication. This finding, and the inferred speciation from a family of plant- and insect-associated viruses suggest that the virus circulates in natural cycles, without published evidence of public health implications.

As part of their ecological interactions warranting virus transmission, several insect species have been incriminated as mechanical or biological vectors. Several insect-specific viruses have been shown to dominate the insect viromes [11], considered essential components of ecosystems [31], and ae able to maintain long-term virus-vector interactions. The successful replication of GUAV in mosquito cells and its inferred ability to infect a natural population of mosquitoes supposes an interaction of the virus with other coinfecting insect-specific viruses and arboviruses, leading to unexpected consequences that deserve study. Additional research on *Culex* spp. vector competence, and virus adaptation to mammalian cells in controlled experimental conditions would help elucidate the molecular and structural basis for receptor binding, host shift, and viral emergence.

## 5. Conclusions

A novel virus, putatively named the *Guachaca virus* (GUAV), was discovered and successfully characterized at molecular and phenotypic levels. This virus was unable to replicate in mammalian cells, which suggests its maintenance as an insect-associated virus. While the GUAV biology and transmission cycle remain unraveled, its natural presence in mosquitoes in rural areas, which are in frequent contact with plants and vertebrates as feeding sources, justifies further studies to unravel the ecological scenario for transmission. Phylogenetic evidence warrants a revision of the current taxonomy of the family *Tymoviridae* to better represent the recently described insect-associated viruses.

## Figures and Tables

**Figure 1 viruses-15-00953-f001:**
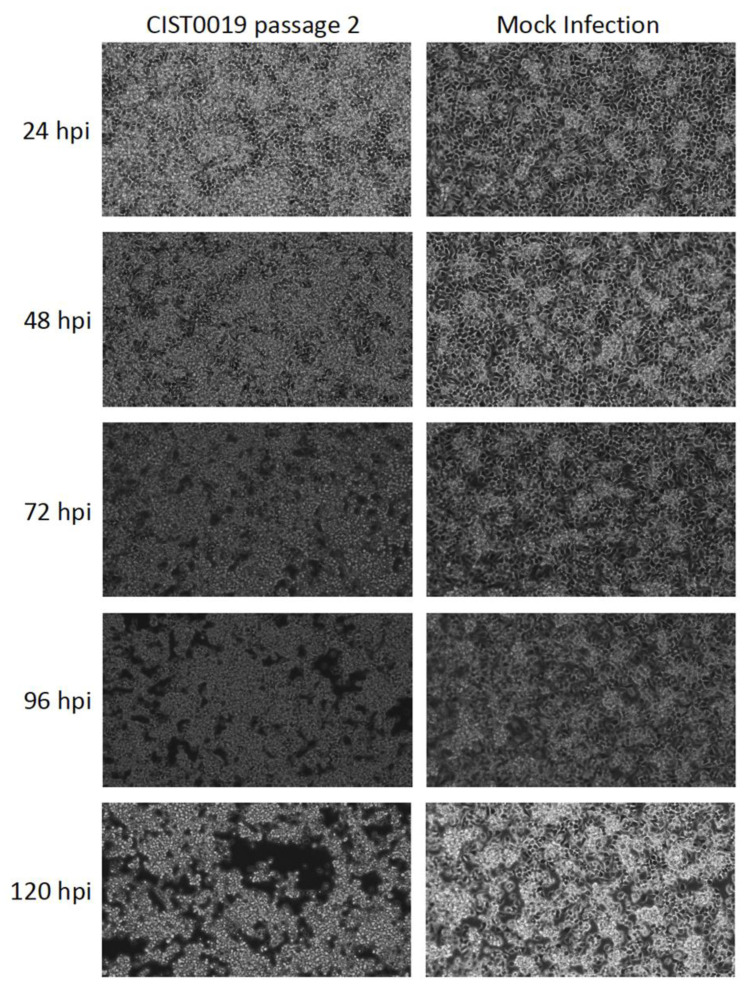
Cytopathic effect induced in C6/36 cells after infection of the sample CIST0019 passage 2. Morphological changes in the cell culture inoculated with the supernatant of the viral infection with mosquito sample CIST0019, characterized by cell aggregation, were observed from the third day post-infection, and increased during the fourth and fifth day. Mock infection: Uninfected C6/36 cells. Magnification: 200×.

**Figure 2 viruses-15-00953-f002:**
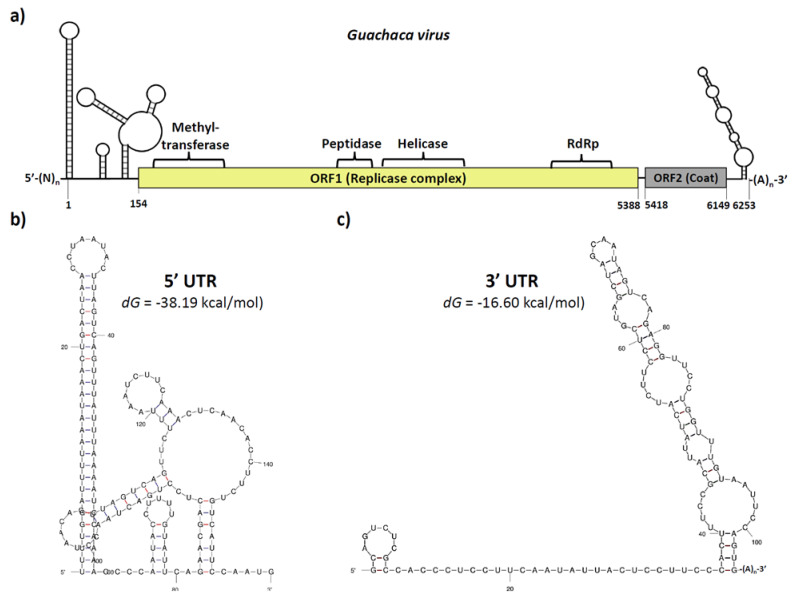
Genomic organization of the putative *Guachaca virus* (GUAV). (**a**) The GUAV genome was approximately 6250 nucleotides in length and comprised a 5′ UTR, a large ORF (ORF1) encoding for the replicase complex, a short ORF (ORF2) encoding for the structural coat protein, a 3′ UTR, and a poly(A) tail. (**b**) The 5′ UTR of around 153 nucleotides in length contains a highly stable IRES-like structure. (**c**) The 3′ UTR of around 125 nt in length contains a long partially mismatched stem-loop structure, followed by a poly(A) tail.

**Figure 3 viruses-15-00953-f003:**
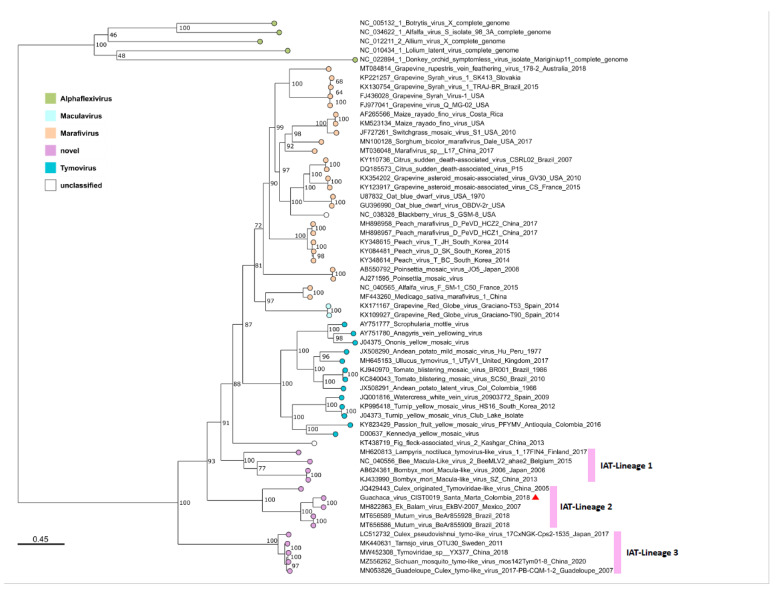
Phylogeny of the family *Tymoviridae*. The tree was reconstructed for a partial replicase complex region (2042 amino acids) of 56 representative sequences of the family *Tymoviridae* and 5 sequences of the family *Alphaflexiviridae* as an outgroup for rooting. The maximum likelihood method was used with the estimated LG + F + I + G4 protein model and 1000 ultrafast bootstrap replicates. Putative GUAV is denoted by a red triangle. Pink bars denote the three insect-associated *Tymoviridae* lineages 1 to 3.

**Figure 4 viruses-15-00953-f004:**
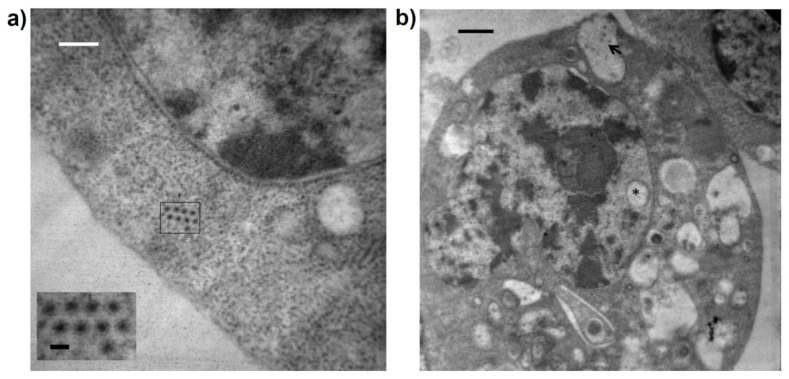
TEM microphotography of C6/36 cells infected with the putative GUAV. (**a**) Complete icosahedral viral particles with a diameter of 40–50 nm were observed in the cytoplasm. Black bar: 50 nm. White bar 700 nm. (**b**) Virus particles observed into vacuoles (arrow). A vacuole was present in the nucleus (asterisk). Black bar 700 nm.

**Table 1 viruses-15-00953-t001:** Primers and probe used for amplification of the viral helicase gene of the putative *Guachaca virus*.

Primer Name	Target Gene	Genomic Position ^a^	Sequence (5′–3′)	Annealing T (°C)
Guachaca-F	Viral helicase	3014–3035	tgacgcccgagacaacttacct	57.5
Guachaca-P	Viral helicase	3067–3090	FAM-acttacaggtctccagccaacatc-BHQ1	55
Guachaca-R	Viral helicase	3164–3142	gagtgacggaaagagcgggagta	58

^a^ Genomic positions were estimated with respect to the ORF1 of the putative *Guachaca virus* based on the genome assembly obtained in the present study (GenBank accession number: OQ286121).

**Table 2 viruses-15-00953-t002:** Estimates of evolutionary divergence within and between genera and suggested lineages belonging to the family *Tymoviridae*.

Group	Amino Acid Differences (d(S.E)) ^a^					
	*Marafivirus*	*Maculavirus*	*Tymovirus*	IAT_Lineage_1	IAT_Lineage_2	IAT_Lineage_3
*Marafivirus*	0.344(0.010)					
*Maculavirus*	0.456(0.013)	0.024(0.005)				
*Tymovirus*	0.500(0.012)	0.515(0.013)	0.381(0.010)			
IST_Lineage_1	0.581(0.013)	0.573(0.014)	0.622(0.013)	0.083(0.006)		
IST_Lineage_2	0.505(0.012)	0.520(0.013)	0.536(0.012)	0.556(0.014)	0.358(0.011)	
IST_Lineage_3	0.536(0.013)	0.534(0.015)	0.555(0.013)	0.602(0.014)	0.540(0.014)	0.254(0.010)

^a^ The number of amino acid differences per site from averaging over all sequence pairs within each group (highlighted in red) and between groups (highlighted in black) are shown. Standard error estimate(s) are shown in square brackets and were obtained by a bootstrap procedure (1000 replicates). This analysis involved 51 amino acid sequences. All ambiguous positions were removed for each sequence pair (pairwise deletion option). The grayscale represents the low-to-high divergence between groups. There were a total of 1160 positions in the final dataset.

## Data Availability

The full-length genome of GUAV has been deposited in GenBank under Accession Number OQ286121.

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
