# Peer review of "Novel Putative Tymoviridae-like Virus Isolated from Culex Mosquitoes in Colombia"

_viruses, 2023, doi:10.3390/v15040953_

Round 1

Reviewer 1 Report

This is a nicely written, descriptive paper detailing the isolation and characterisation of a novel virus from Culex mosquitoes in Colombia. This paper adds to our growing knowledge of the depth of insect viromes and highlights the importance of understanding not just the vector-animal host interaction but the interaction with mosquito's other food source - plants.

My overall comments:

- The introduction reads very well and covers all the relevant background for this paper.

- The methods are well described and contain all necessary information.

- The results are clear and the figures are well put together. I particularly like the work that has been done to annotate the genome so thoroughly. The article is appropriately referenced.

- My only issue is with the link the authors appear to make between discovery of these Tymo-like viruses and their potential relevance to mammalian hosts due to the feeding preferences of Culex mosquitoes (lines 70-71, 349 and 402-403) . I'm not convinced that a relative of plant-infecting viruses poses a particularly high risk of adapting to an animal host, if this is indeed what the authors are suggesting. However, I do not think this diminishes the importance of this paper and the discovery of this group of viruses which are incredibly interesting in their own right.

- In descriptive papers like these physical characterisation such as SDS-PAGE analysis of virion proteins can be very useful, particularly for more novel viruses where limited information is available. I would recommend including this data if you have it as I think it would provide a valuable resource for others studying these viruses, but I am happy for the paper to be published without it.

- Congratulations on a very nice paper!

Specific comments:

ABSTRACT

Line 23 - Typo in this sentence add an "and"

INTRODUCTION

line 70-71 - It's not clear exactly what the point of this statement is. I would recommend removing it or rewording it to be clearer.

METHODS

Line 81 - this sentence appears to be incomplete
Line 190 - change "y" to "and"

RESULTS

Line 239 - should read "59.4%" rather than "59,4,"
Figure S1 - there is no mention of figure S1 B in the text, it would be good to include some description of the coat protein domains that have been annotated
Figure S2 - not sure it is necessary to show the gel for RACE, also the original gel image does not look like the original image since it is the same as the figure image and is cropped
The authors have done an excellent job annotating the genome and figure 2 is beautiful.
Line 305-308 - What is the difference between IST-lineage and IAT-lineage? If they are referring to the same thing I suggest using one term only for consistency

DISCUSSION

Line 349 - Its not clear what point the authors are trying to make with this sentence, I would suggest rewording to make it clearer or removing
Line 365 - authors say it is of interest that a stem loop structure was identified in the 3' end of the genome - please expand on how this relates to other members of the order or not
Line 390 - This statement is a bit of a stretch. It would be more of interest to focus on how this virus might interact with other viruses circulating in these mosquito populations (potential for super-infection exclusion), how these viruses might impact the mosquito itself (host-fitness, as investigated for negeviruses) or even the potential impact of these viruses on plants.

CONCLUSION

I think the conclusion about the virus likely being maintained in insect-specific cycle is sound, and the authors present a strong case for revision of the Tymoviridae family taxonomy. However I do not agree with the statement that discovery of this virus and the feeding preference of its host is a reason for arbovirus surveillance in the area it was discovered.
I think a stronger argument would be that the discovery of this novel virus warrants further investigation of the virome of mosquitoes in this region, highlights the importance of virus discovery, etc.

Note: Institutional Review Board Statement is not filled out yet

Reviewer 2 Report

#minor

110: Authors describe maintenance of Vero cells at 28C (lines 99-102) but infect cells at 37C? Is this information correct?

114: “The virus isolation supernatant was collected when the cytopathic effect was evidenced”. Enter how long dpi the cytopathic effect was evidenced

120-121: “and the cell culture was incubated until the cytopathic effect was observed in more than 50% of the cell monolayer” Enter how long dpi the cytopathic effect was evidenced

189-190: “5 ml of fresh culture medium added”. Enter the percentage of FBS in the maintenance medium?

222-223: replace “days of/post inoculation” by “days post infection”. Review the entire manuscript and make the appropriate replacement

369: “phylogenetic analysis presented here and some previously reported studies” Insert appropriate references to studies cited

384-385: “This finding and the inferred speciation of a family of viruses associated with plants and insects suggest that the virus circulates in natural cycles without public health implications” Which references support this statment?

389-392: “and being able to maintain long-term virus-vector interactions. The successful replication of RNA viruses in mosquitoes, and their implicit adaptive potential for successful exploitation of novel environments, could lead to unexpected consequences and potential emergence if a contact of these viruses with mammalian cells is sustained through time” . Insert the appropriate references to the information presented
